# Critical Role of Novel O-GlcNAcylation of S550 and S551 on the p65 Subunit of NF-κB in Pancreatic Cancer

**DOI:** 10.3390/cancers15194742

**Published:** 2023-09-27

**Authors:** Aishat Motolani, Matthew Martin, Benlian Wang, Guanglong Jiang, Faranak Alipourgivi, Xiumei Huang, Ahmad Safa, Yunlong Liu, Tao Lu

**Affiliations:** 1Department of Pharmacology and Toxicology, Indiana University School of Medicine, 635 Barnhill Drive, Indianapolis, IN 46202, USA; amotolan@iu.edu (A.M.); marti336@gmail.com (M.M.); asafa@iupui.edu (A.S.); 2Center for Proteomics and Bioinformatics, Case Western Reserve University, Cleveland, OH 44106, USA; bwang@wakehealth.edu; 3Department of Medical & Molecular Genetics, Indiana University School of Medicine, Indianapolis, IN 46202, USA; ggjiang@iu.edu (G.J.); yunliu@iu.edu (Y.L.); 4Melvin and Bren Simon Comprehensive Cancer Center, Indiana University School of Medicine, Indianapolis, IN 46202, USA; falipour@iu.edu (F.A.); xiuhuang@iu.edu (X.H.); 5Department of Radiation Oncology, Indiana University School of Medicine, Indianapolis, IN 46202, USA; 6Department of Biochemistry and Molecular Biology, Indiana University School of Medicine, 635 Barnhill Drive, Indianapolis, IN 46202, USA

**Keywords:** NF-κB, O-GlcNAcylation, pancreatic cancer, post-translational modification

## Abstract

**Simple Summary:**

NF-κB is an inflammatory protein that contributes to the low rate of survival observed in pancreatic cancer patients. The activation of NF-κB causes high expression of proteins that drive and sustain pancreatic cancer growth. Thus, it is important to understand how NF-κB is regulated to help with the diagnosis and treatment of pancreatic cancer. In this study, we discovered that the typical NF-κB subunit p65 is modified by O-GlcNAc at serines (S)550 and S551. To characterize the role of O-GlcNAcylated p65 at S550 and S551, we overexpressed p65 serine-to-alanine (S-A) mutants, such as S550A, S551A, and S550A/S551A, in pancreatic cancer cells. Using this model, we show that the p65 mutants reduce NF-κB transcriptional activity, nuclear translocation, p65 phosphorylation, and target gene expression. We also observed that the p65 mutants blocked pancreatic cancer cell growth and migration. This suggests the contribution of p65 O-GlcNAcylation at S550 and S551 to pancreatic cancer phenotypes. Taken together, our study uncovers a novel aspect of NF-κB regulation, which could aid future therapeutic development by targeting O-GlcNAc transferase (OGT) in pancreatic cancer.

**Abstract:**

Pancreatic ductal adenocarcinoma (PDAC) is one of the most lethal malignancies, with a mere 5-year survival of ~10%. This highlights the urgent need for innovative treatment options for PDAC patients. The nuclear factor κB (NF-κB) is a crucial transcription factor that is constitutively activated in PDAC. It mediates the transcription of oncogenic and inflammatory genes that facilitate multiple PDAC phenotypes. Thus, a better understanding of the mechanistic underpinnings of NF-κB activation holds great promise for PDAC diagnosis and effective therapeutics. Here, we report a novel finding that the p65 subunit of NF-κB is O-GlcNAcylated at serine 550 and 551 upon NF-κB activation. Importantly, the overexpression of either serine-to-alanine (S-A) single mutant (S550A or S551A) or double mutant (S550A/S551A) of p65 in PDAC cells impaired NF-κB nuclear translocation, p65 phosphorylation, and transcriptional activity, independent of IκBα degradation. Moreover, the p65 mutants downregulate a category of NF-κB-target genes, which play a role in perpetuating major cancer hallmarks. We further show that overexpression of the p65 mutants inhibited cellular proliferation, migration, and anchorage-independent growth of PDAC cells compared to WT-p65. Collectively, we discovered novel serine sites of p65 O-GlcNAcylation that drive NF-κB activation and PDAC phenotypes, thus opening new avenues by inhibiting the NF-κB O-GlcNAcylation enzyme, O-GlcNAc transferase (OGT), for PDAC treatment in the future.

## 1. Introduction

Pancreatic Ductal Adenocarcinoma (PDAC), which accounts for over 90% of pancreatic cancer cases, is experiencing a rapid increase in incidence. Its overall five-year survival rate stands at a dismal 12% [1]. This poor prognosis of PDAC can be attributed to factors such as late clinical manifestation, limited availability of effective therapeutics, and a lack of early diagnostic biomarkers, among others [2]. Current treatment options for PDAC patients include surgical resection and chemotherapy [3]. Despite these treatments, patients often face poor outcomes, including tumor relapse or death. This underscores the urgent need for the discovery of new targets and the development of novel therapeutic strategies. One particularly promising target is the nuclear factor kappa B (NF-κB) signaling pathway.

NF-κB is a master regulator of inflammation known to be constitutively active in PDAC [4]. This ubiquitous transcription factor is composed of five subunits (RelA/p65, RelB, c-Rel, NF-κB1 (p50 and p105), and NF-κB2 (p52 and p100)) that dimerize to control gene expression. The canonical NF-κB pathway is activated by various stimuli such as cytokines, growth factors, radiation, or stress, leading to phosphorylation of IKKβ (IκB kinase β) and phosphorylation-induced degradation of IκBα (NF-κB inhibitor α). Then, the p65–p50 dimers translocate into the nucleus to mediate transcription of NF-κB target genes [5].

Notably, in PDAC, aberrant signaling of the canonical NF-κB pathway contributes to various cancer hallmarks, such as metastasis, chemoresistance, cell proliferation, and cell survival, among others [4]. Specifically, the dysregulation of the p65 subunit of NF-κB through post-translational modifications (PTMs) plays a significant role in cancer progression [6,7]. For instance, our research group and others have demonstrated that the deregulation of PTMs, such as methylation, phosphorylation, and acetylation, on p65 or its regulators critically influences the development, progression, and aggressiveness of several cancers, including PDAC [5,8,9,10]. Given the extensive role of PTMs in dynamically regulating p65, our aim is to identify novel PTMs on p65 to advance our understanding of NF-κB signaling, particularly in the context of PDAC.

O-GlcNAcylation involves the transfer of an N-acetylglucosamine (GlcNAc) molecule from the donor molecule, UDP-GlcNAc, to an acceptor protein’s serine or threonine residue by O-linked N-acetyl-glucosaminyl transferase (OGT). This modification can be reversed by the O-linked N-acetyl-glycosaminidase (OGA) [11]. Elevated O-GlcNAcylation of proteins in cells is a common feature of cancers such as PDAC, lung, ovarian, gastric, prostate, and breast cancers [12]. Interestingly, the hyper-O-GlcNAcylation in PDAC has been shown to promote tumor cell survival and increase NF-κB signaling [13]. Given the critical role that the p65 transactivation domain (TAD) (amino acids 428–551) plays in facilitating most of NF-κB activity [14], it is important to understand if O-GlcNAcylation directly regulates the function of this domain.

In this study, we have uncovered O-GlcNAcylation of p65 at S550 and S551, both located within the TAD domain of p65. We further elucidate the mechanistic function of these termini serine residues by showing that overexpression of S550A and/or S551A p65 mutants leads to significantly reduced NF-κB transcriptional activity, NF-κB target gene expression, and nuclear localization in PDAC cell lines. Moreover, p65 mutants significantly decrease PDAC cellular proliferation, migration, and anchorage-independent growth. Collectively, our findings unveil novel O-GlcNAcylated sites on p65 and their consequential effects on NF-κB signaling and PDAC phenotypes. This discovery of functional O-GlcNAcylated sites on the NF-κB protein holds promise for advancing the development of effective PDAC therapies targeting the OGT/NF-κB axis.

## 2. Materials and Methods

### 2.1. Liquid Chromatography-Tandem Mass Spectrometric Analysis

A coomassie-stained SDS-PAGE gel band containing FLAG-tagged WT-p65 (FLAG-WT-p65) protein was subjected to in-gel tryptic digestion. FLAG-WT-p65 gel pieces were subjected to destaining and reduction of cysteine residues using 50% acetonitrile in 100 mM ammonium bicarbonate and 100% acetonitrile, followed by treatment, followed by 20 mM DTT at room temperature for 60 min. Alkylation with 55 mM iodoacetamide for 30 min was performed in the dark. The solution was removed, and the gel pieces were washed with 100 mM ammonium bicarbonate and dehydrated in acetonitrile. Gel pieces were then dried in a SpeedVac centrifuge and proteolytically digested by rehydration overnight at 37 °C in 50 mM ammonium bicarbonate containing sequencing grade modified trypsin (Promega, WI, USA). Extracted peptides were treated with 50% acetonitrile in 5% formic acid, dried, and reconstituted in 0.1% formic acid for mass spectrometry analysis. Analysis of proteolytic digests was performed by using an LTQ Orbitrap XL linear ion-trap mass spectrometer (Thermo Fisher Scientific, Waltham, MA, USA), coupled with an Ultimate 3000 HPLC system (Dionex, Sunnyvale, CA, USA). The digests were injected onto a reverse-phase C18 column (0.075 × 150 mm, Dionex) equilibrated with 0.1% formic acid/4% acetonitrile (*vol*/*vol*). A linear gradient of acetonitrile from 4 to 40% in water in the presence of 0.1% formic acid over a period of 45 min was used at a flow rate of 300 nL/min. The spectra were acquired by data-dependent methods, consisting of a full scan (m/z 400–2000) and then tandems on the five most abundant precursor ions. The previously selected precursor ions were scanned once during 30 s and then excluded for 30 s. The obtained data were analyzed by Mascot software (Matrix Science, Horsham, PA, USA) against a customized p65 protein database with a setting of 10 ppm for precursor ions and 0.8 Da for product ions. Carbamidomethylation of cysteine was set as a fixed modification, while oxidation of methionine, O-GlcNAcylation of serine, and threonine were set as variable modifications. The tandem mass spectra of candidate-modified peptides were further interpreted manually.

### 2.2. Cell Lines and Transfections

PDAC cell lines PANC1 and MIA PaCa2 were purchased from ATCC (Manassas, VA, USA) and cultured in Cytiva HyClone™ Dulbecco’s High Glucose Modified Eagles Medium (DMEM) supplemented with 5 mL of 10,000 units/mL penicillin, 10,000 units/mL streptomycin, and 5% fetal bovine serum (FBS). The FLAG-S550A, S551A, and S550A/S551A mutants of p65 were generated with the QuikChange II XL Site-Directed Mutagenesis Kit from Agilent Technologies. Primers were designed using the Agilent Technologies QuikChange Primer Design online software. Constructs were transfected into PDAC cell lines as described previously [8] using Lipofectamine (Thermo Fisher Scientific, Waltham, MA, USA).

### 2.3. Western Blotting and Antibodies

Cells were cultured to about 90–95% confluence before lysis. Whole cell samples were collected and lysed using Radio Immunoprecipitation Assay buffer (RIPA buffer: 150 mM NaCl, 0.1% Triton X-100, 0.5% sodium deoxycholate, 0.1% sodium dodecyl sulfate (SDS), 50 mM Tris-HCl pH 8.0, and protease inhibitors). Whole cell lysates were separated by SDS/PAGE gels, and proteins were probed by Western blotting. Antibodies used to detect the target proteins include anti-p65 (Santa Cruz Biotechnology, sc-372, Dallas, TX, USA), anti-FLAG (Sigma-Aldrich, F1804, St. Louis, MI, USA), anti-O-GlcNAc (Santa Cruz Biotechnology, sc-59623, Dallas, TX, USA), anti-IκBα (Cell signaling technology, 9242s, Dallas, TX, USA), anti-pS536 (Cell signaling, ab3031, Danvers, MA, USA), and anti-beta actin (Sigma-Aldrich, A5316, St. Louis, MO, USA). Quantifica-tions were performed using ImageJ on three inde-pendent western blot images.

### 2.4. Immunofluorescence

1 × 10^5^ PANC1 or MIA PaCa2 cells were seeded onto coverslips in a 24-well plate. The next day, the cells were treated with or without IL-1β for 1 h. After treatment, the cells were fixed with 4% formaldehyde for 30 min, and the reaction was quenched by 100 mM glycine for 5 min. The cells were gently washed with 1× PBS and then blocked and permeabilized with blocking buffer (1XPBST and 1% BSA) and permeabilizing buffer (Blocking buffer with 0.2% Tritonx-100), respectively. Cells were further probed with anti-FLAG antibodies for FLAG-tagged WT-p65 or S550A, S551A, and S550A/S551A and Alexa Fluor 488 (green) goat antimouse IgG. Before sealing the coverslips, mounting medium with DAPI was used to stain the nucleus. The slides were examined under a Leica DMI6000B series fluorescent microscope at 40× magnification.

### 2.5. Luciferase Assays

NF-κB luciferase assays were conducted by transiently transfecting the κB-luciferase construct ELAM-luc into parental control or FLAG-WT-p65 stable cell lines. Cells were treated with or without OSMI-1 (Cayman chemical, 21894-1, Ann Arbor, MI, USA) for 24 h. Luciferase activity was quantified 48 h after transfection using the Luciferase Assay System with Reporter Lysis Buffer kit (Promega, Fitchburg, WI, USA). The ELAM-luc plasmid contains three NF-κB binding sites derived from the E-selectin gene located upstream of a luciferase reporter gene. A β-galactosidase construct was used to normalize for transfection efficiency. Luciferase activity was measured using a Synergy H1 Multi-Mode Reader (BioTek Instruments Inc., Winooski, VT, USA).

### 2.6. Co-Immunoprecipitations

Cells stably expressing FLAG-tagged p65 proteins were cultured to ~80% confluency and then treated with Thiamet G (Santa Cruz Biotechnology, sc-224307A) for 24 h. Following treatment, cells were lysed in co-immunoprecipitation buffer (1% Triton X-100 (*v*/*v*), 50 mM Tris-HCl (pH 7.4), 150 mM NaCl, 1 mM EDTA, 1 mM sodium orthovanadate, 20 µM aprotinin, 1 mM phenylmethanesulfonyl fluoride, and 1 mM pepstatin A). FLAG-tagged p65 proteins were then immunoprecipitated with anti-FLAG-M2 beads (Sigma-Aldrich, St. Louis, MO, USA), using immunoprecipitation methods previously described [9]. Briefly, cell lysates with equivalent amounts of protein were incubated with anti-FLAG-M2 beads at 4 °C overnight. Beads were then washed, and FLAG-tagged proteins were eluted and separated by SDS/PAGE [15].

### 2.7. Cell Proliferation and 3D Growth Assays

For cell proliferation assays, PDAC-stable cells overexpressing FLAG-WT-p65 and mutant p65 constructs were seeded in triplicate at 2 × 10^4^ cells/well in a 6-well plate. PANC1 cells were counted at days 3, 5, 7, 9, and 11 days, while MIA PaCa2 cells were counted at days 3, 5, 7, and 9 after seeding using a cell counting chamber. For 3D growth assays, Matrigel (Fisher scientific, Waltham, MA, USA) was used to prepare 3% Matrigel and media cell suspension. 250 cells were seeded and cultured for 5 days at 37 °C and 5% CO_2_. On day 5, formed spheroids were imaged and captured using a Canon EOS Rebel T3i Digital SLR camera and treated with Alamar Blue (Sigma-Aldrich, St. Louis, MO, USA) at 10% culture volume. Fluorescence intensity was read using the Synergy H1 Multi-Mode Reader (BioTek Instruments Inc., Winooski, VT, USA) at an excitation wavelength of 544 nm and an emission wavelength of 590 nm.

### 2.8. Migration Assay

Migration assays were conducted using Boyden chambers. Briefly, a Boyden chamber consists of 8 μm pore size cell culture inserts in a 24-well plate. Each insert was coated with gelatin on the side facing the lower chamber. In serum-free media, 2 × 10^5^ cells were seeded in the top of the insert (upper chamber), while serum-rich media (10% serum) was supplied in the well below as a chemoattractant. After 24 h, migrated cells were fixed with 4% formaldehyde and stained with crystal violet. Stained cells were visualized with a light microscope and quantified. Images were captured using a Canon EOS Rebel T3i Digital SLR camera. Quantification was performed using imageJ.

### 2.9. Quantitative PCR (qPCR)

qPCR experiments were carried out as previously described (6). Briefly, established PDAC stable cells were cultured to ~90% confluence, and total RNA was isolated using Trizol reagent (Invitrogen, Carlsbad, CA, USA). Total isolated RNA was used to prepare cDNA using the SuperScript III First-Strand Synthesis PCR System (Invitrogen, Carlsbad, CA, USA). qPCR was carried out using FastStart Universal SYBR Green Master ROX (Roche, Basel, Switzerland). Primers were designed by the NCBI Primer BLAST tool. Primer sequence is as shown: CD274-Forward: 5′-GGTGCCGACTACAAGCGAAT-3′; CD274-Reverse: 5′-TGACTGGATCCACAACCAAAATT-3′; FGF8-Forward: 5′-AGTACCGACCCGCACGCTCTT-3′; FGF8-Reverse: 5′-GACCAGCAAGTGCAACAGCCACG-3′.

### 2.10. Ingenuity Pathway Analysis (IPA)

A list of genes downregulated by S550A, S551A, and S550A/S551A was analyzed by the IPA software. The setting and filter were as follows: reference set: ingenuity knowledge base (Genes_Endogenous Chemicals); relationship to include: direct and indirect; includes endogenous chemicals; filter summary: consider only molecules where species were human, rat, or mouse.

## 3. Results

### 3.1. The p65 Subunit of NF-κB Is O-GlcNAcylated in the Transcriptional Activating Domain at S550 and S551

NF-κB is subjected to several PTMs that fine-tune its function. This phenomenon highlights one of the complexities of the NF-κB signaling pathway in cancer [16]. Thus, to expand our understanding of the NF-κB pathway, using the tandem mass spectrometry technique, we scanned for PTMs on a p65 peptide isolated from a HEK 293 cell-derived cell line with constitutive NF-κB activity [17]. As shown in Figure 1A, we discovered that the terminal serine sites of p65, S550, and S551 are modified by the N-acetylglucosamine (GlcNAc) molecule. Notably, O-GlcNAcylation is critical to the malignant processes that drive pancreatic cancer [13]. Thus, we decided to evaluate the regulatory role of the newly discovered p65 GlcNAc sites in PDAC. First, we generated PANC1 and MIA PaCa2 cell lines stably expressing empty lentiviral vector, FLAG-tagged wild-type p65 (FLAG-WT-p65), and mutant p65 (FLAG-S550A, FLAG-S551A, and FLAG-S550A/S551A) (Figure 1B). Following the establishment of PDAC stable cells, we validated the mass spectrometry data by immunoprecipitating FLAG-WT-p65 and mutant p65 from the PDAC cells and assessing their GlcNAc levels. As shown in Figure 1C, FLAG-WT-p65 is significantly O-GlcNAcylated compared to FLAG-S550A, -S551A, and -S550A/S551A, indicating that S550 and S551 constitute major O-GlcNAcylation sites in PDAC cells. In addition, the YinOYang server [18], known for predicting O-GlcNAc sites on intracellular and nuclear proteins, listed S550 and S551 sites on p65 with a high potential to be O-GlcNAcylated, surpassing the O-GlcNAc potential of sites identified in the N-terminal domain, some of which have been reported to exert no effect on NF-κB activity. (Figure 1D). Notably, both S550 and S551 residues of p65 are highly conserved across different species, as indicated by the cross-species alignment in Figure 1E, suggesting a potentially important function of these sites.

### 3.2. O-GlcNAcylation of p65 at S550 and S551 Is Important for NF-κB Transcriptional Activity

The constitutive activation of NF-κB transcriptional activity enhances PDAC progression [22]. Previous studies have demonstrated that site-specific O-GlcNAcylation of p65 regulates its transcriptional activity [13,19,20]. Thus, considering that S550 and S551 are in the TAD domain of p65, we wondered if p65 mutants impact NF-κB transcriptional function. Herein, we performed an NF-κB luciferase activity assay with or without OSMI-1 treatment. OSMI-1 is a potent inhibitor of OGT’s enzymatic activity and reduces O-GlcNAcylation of proteins [23]. As shown in Figure 2, p65 mutant cells exhibit significantly lower NF-κB activity than WT-p65 in both PANC1 (Figure 2A) and MIA PaCa2 cells (Figure 2B). Particularly, we observed the highest reduction of NF-κB activity in cells expressing the S550A/S551A double mutant, suggesting the critical importance of O-GlcNAcylation of both serine sites. To further confirm if the transcriptional changes caused by p65 mutants are due to the O-GlcNAcylation of S550 and S551, we treated the PANC1 and MIA PaCa2 stable cells with OSMI-1 (Figure 2A,B). Herein, we observed reduced NF-κB activity in WT-p65 cells compared to its control. Meanwhile, in p65 mutants, we observed a much lower decrease in the treated cells compared to the untreated cells, suggesting that the O-GlcNAcylation of p65 S550 and S551 play a key role in NF-κB transcriptional function.

### 3.3. The S550A and S551A mutations of p65 Downregulate a Subset of NF-κB Target Genes

The regulation of p65 by PTMs also results in differential gene expression [10]. Currently, there is no literature report on the role of p65 O-GlcNAcylation on differential gene expression in PDAC. Hence, we aim to elucidate the impact of S550A, S551A, or S550A/S551A p65 mutants on gene expression levels in PDAC cells. As a result, we performed RNA-sequencing (RNA-seq) on our established PANC1 stable cell lines (Figure 1B). Our data suggested that 986 genes were upregulated by 2.5-fold or more in WT-p65 cells than in vector cells. Similarly, out of the 986 genes, 71.6%, 68.2%, and 73.1% were significantly downregulated in S550A, S551A, and S550A/S551A cells, respectively (Figure 3A). We wondered if the simultaneous presence of two serine O-GlcNAcylation on p65 is unique or redundant in their control of gene expression. To assess this question, we analyzed the pool of downregulated genes in all p65 mutants. We reveal that S550A and S551A commonly downregulate ~88–92% of genes, while all mutants commonly downregulate ~50% of genes (Figure 3B). This indicates that S550 and S551 O-GlcNAcylation control the majority of NF-κB-dependent gene expression in PDAC cells. Furthermore, ingenuity pathway analysis (IPA) revealed the top regulators of the commonly downregulated genes in p65 mutant cells. These include the NF-κB complex, p50 subunit, interferon-gamma (IFN-γ), tumor necrosis factor (TNF), B-Cell Lymphoma 3 (Bcl3), interleukin 17A (IL-17A), etc. (Figure 3C). Similarly, the genes downregulated in p65 mutant cells are significantly enriched in several oncogenic cellular and biological functions, such as cellular development, molecular mechanisms of cancer, cell death/survival, tumor microenvironment pathways, etc. (Figure 3C). In Figure 3D, we show that many of the genes downregulated in p65 mutant cells interact within a network controlled by the NF-κB complex. Interestingly, some of the commonly downregulated genes in the p65 mutant cells include immune and proinflammatory factors and their receptors, and cell cycle proteins, such as CD274 (Programmed cell death 1 ligand 1, PD-L1), CDKL3 (Cyclin-dependent kinase-like 3), FGF8 (Fibroblast growth factor 8), GZMM (Granzyme M), HDAC9 (Histone deacetylase 9), IL-15, IL-16, IL23A (IL23 subunit α), IL20RB (IL20 receptor subunit β), and TNFSF13 (TNF superfamily member 13), etc. (Figure 3E). To validate a few of the gene expressions shown in Figure 3F, we assessed the transcript levels of CD274 (PD-L1) and FGF8 by qPCR in PANC1 and MIA PaCa2 stable cells. We observed that CD274 (PD-L1) and FGF8 transcripts are significantly downregulated in p65 mutant cells compared to WT-p65 cells (Figure 3E). Taken together, these data suggest that O-GlcNAcylation of p65 at S550 and S551 is critical for the expression of a subset of NF-κB target genes that play an important role in inflammatory responses and cancer-associated hallmarks.

### 3.4. The S550A and S551A Mutations of p65 Reduce NF-ĸB Nuclear Translocation and Does Not Affect IĸBα Degradation

The global elevation of cellular O-GlcNAcylation has been reported to increase the nuclear accumulation of p65 in murine embryonic fibroblasts (MEFs) [19]. Considering that the p65 mutants (S550A, S551A, and S550A/S551A) have demonstrated a clear reduction in NF-ĸB transcriptional activity and, consequently, gene expression, we sought to investigate the underlying mechanisms driving this observed outcome. Hence, we assessed the impact of p65 mutants on nuclear localization and IĸBα degradation in PDAC cells. To determine FLAG-tagged p65 nuclear translocation, we performed immunofluorescence experiments using an anti-FLAG antibody and DAPI. As shown in Figure 4A,B, we observed significantly increased p65 nuclear translocation in WT-p65 cells compared to S550A, S551A, and S550A/S551A cells upon NF-ĸB activation with IL-1β for 1 h. In summary, our data strongly suggest that O-GlcNAcylation of p65 at S550 and S551 influences p65 nuclear translocation. Furthermore, we investigated the pattern of IκBα degradation through Western blot analysis in both PANC1 and MIA PaCa2 stable cell lines. As illustrated in Figure 4C, we observed no significant difference in IκBα degradation in both WT-p65 and mutant p65 PANC1 cell lines. In Figure 4D, similar experiments were conducted in MIA PaCa2 stable cells, and no significant difference in IκBα degradation pattern was observed between WT-p65 and p65 mutant cells. Notably, the IκBα degradation pattern observed in MIA PaCa2 appeared to be slower and subtle compared to that of PANC1, possibly attributable to differences in cell type. Overall, our data suggests that the O-GlcNAcylation of S550 and S551 does not exert a significant influence on IκBα degradation in PDAC cells.

### 3.5. S550A and S551A Mutations of p65 May Compromise S536 Phosphorylation in PDAC

Numerous studies have highlighted the intricate interplay between O-GlcNAcylation and phosphorylation in proteins [24]. Notably, the protein p65 undergoes extensive phosphorylation at various serine and threonine residues, influencing NF-κB transcriptional activity, stability, and interactions with other molecules [9,25]. One particular site of interest, p65 S536, is subject to phosphorylation by several kinases, including IKKα/β/ε, Ribosomal Subunit S6 Kinase 1 (RSK1), and NF-κB activating kinase (NAK)/TANK-binding kinase 1 (TBK1) [25]. Previously, our laboratory demonstrated the dynamic phosphorylation of S536 in response to IL-1β treatment in 293C6 cells [9]. Similarly, Ma and colleagues reported a significant reduction in S536 phosphorylation in MIA Pa-Ca2 and BxPC-3 cells following OGT knockdown [13]. However, the extent to which this decrease is attributed to the p65 O-GlcNAc modification at S550 and S551 remains unclear. Our experimental findings, as illustrated in Figure 5A,B, reveal a generally more prolonged p65 phosphorylation pattern in WT-p65 PDAC cells upon IL-1 treatment, compared to p65 mutant cells. These findings suggest the potential involvement of O-GlcNAcylation at S550 and S551 in modulating S536 phosphorylation in PDAC cells.

### 3.6. The S550A and S551A Mutations of p65 Decreases Cell Proliferation, 3D Growth, and Cell Migration in PDAC

Our IPA findings revealed a noteworthy observation: numerous genes downregulated by S550A and/or S551A are intricately linked to cancer-related functions (Figure 3C). In light of this discovery, we investigated the impact of p65 mutations (S550A, S551A, and S550A/S551A) on pivotal cancer phenotypes, including cellular proliferation, anchorage-independent growth, and migration. First, we cultured PANC1 and MIA PaCa2 stable cells to examine their cell growth rates. As illustrated in Figure 6A, we showed that p65 mutant cells displayed a significant decrease in growth before reaching plateau compared to WT-p65 cells. In addition, anchorage-independent growth in tumor cells plays a critical role in promoting metastasis and facilitating the expansion and invasion of tumor cells into nearby or distant tissues [26]. Thus, we wondered whether S550A and/or S551A impacted the ability of the cells to grow in an anchorage-independent manner. After culturing the PDAC stable cells in 3D for five days, we treated the cells with Alarma blue to quantify the anchorage-independent cell growth. As shown in Figure 6B, the assay revealed significantly increased spheroid formation in cells expressing WT-p65 compared to those expressing p65 mutants. To gain additional insights into the functional implications of these newly identified O-GlcNAc sites, we evaluated the role of S550A, S551A, and S550A/S551A in modulating PDAC cell migration. Notably, the overexpression of p65 mutants, in comparison to WT-p65, yielded a significant reduction in the number of migrated cells, as depicted in Figure 6C. In conclusion, our data collectively underscores the significance of p65 O-GlcNAcylation at S550 and S551 in regulating the proliferation, anchorage-independent growth, and migration of PDAC cells.

### 3.7. OGT Protein Abundance Positively Correlates with p65 and Is Elevated in PDAC Patients

O-GlcNAcylation is regulated by only one pair of enzymes, namely OGT and OGA [27]. Interestingly, previous studies have reported increased OGT activity in PDAC cells and elevated OGT expression in human pancreatic tissue microarrays up to stage 3 [13,28]. In our present study, using a dataset obtained from the Clinical Proteomic Tumor Analysis Consortium (CPTAC), we found that OGT and p65 protein levels are indeed positively correlated (Figure 7A). Moreover, in concordance with previous studies, using the CPTAC PDAC dataset, we observed that OGT proteomic expression increases as PDAC evolves from stage 1 to stage 4 (Figure 7B), suggesting the potential combined clinical significance of OGT and p65 to the progression of PDAC.

### 3.8. Hypothetical Model

As proposed in Figure 8, in this study, we provide evidence that p65 is O-GlcNAcylated at S550 and S551, both of which are critical residues in the transactivation domain of p65. Mechanistically, the O-GlcNAcylation of p65 at S550 and S551 may promote NF-κB nuclear translocation, enhance the strength and duration of p65 phosphorylation on S536, and increase the transcriptional activity independent of IκBα degradation, leading to the differential expression of critical NF-κB target genes such as CD274 (PD-L1), FGF8, IL-15, IL-16, HDAC9, and so on. Concurrently, these gene expression changes may affect multiple cancer phenotypes driven by NF-κB activation, such as proliferation, anchorage-independent growth, and migration in PDAC.

## 4. Discussion

NF-κB is a well-established link between inflammation and cancer, including PDAC. The aberrant signaling of NF-κB in PDAC is driven by different factors, such as PTMs, which adds a significant layer of complexity to the outcomes of NF-κB signaling [29]. Notably, O-GlcNAcylation is an emerging PTM critical for NF-κB regulation in cancer. In this study, we became the first to uncover that the p65 subunit of NF-κB is O-GlcNAcylated at S550 and S551 in PDAC cells (Figure 1). Furthermore, our findings indicate that the O-GlcNAcylation of p65 at S550 and S551 is critical for NF-κB transcriptional activity, nuclear translocation, p65 S536 phosphorylation, differential gene expression, cellular proliferation, anchorage-independent growth, migration, etc., in PDAC (Figure 2, Figure 3, Figure 4, Figure 5 and Figure 6).

Since its discovery in 1983, O-GlcNAcylation has been reported to modify about 4000 proteins, including inflammatory proteins such as NF-κB, IKKβ, STAT3 (Signal transducer and activator of transcription 3), TAK-1, NFAT (Nuclear factor of activated T cells), Sp1, etc. [30,31]. Thus, the O-GlcNAc modification alters various cellular processes through increased protein activity, protein stability, protein-protein interaction, and protein subcellular localization [30]. OGT has been reported to regulate NF-κB and is required for NF-κB activation and inflammation in different diseases, such as pancreatitis [32] and lung metastasis [33]. In another report, Phoomak et al. reported that O-GlcNAcylation of p65 promotes its nuclear translocation, leading to high metalloproteinase 7 (MMP7) expression, which drives increased migration and invasion of cholangiocarcinoma cells [34].

Interestingly, the connection between high blood glucose, O-GlcNAcylation, and proinflammatory cytokine production has also been documented. For instance, Ferrer et al. suggested that O-GlcNAcylation controls major metabolic and signaling pathways known to promote cancer hallmarks [12]. Cancer cells have an increased dependence on glucose and glutamine for oxygen-independent metabolism [21]. Martinez-Outschoorn et al. suggested that a fraction of the total glucose and glutamine entering the cell goes into the hexosamine biosynthetic pathway (HBP), which produces the donor molecule UDP-GlcNAc for the formation of O-GlcNAcylation on protein substrates [21]. Additionally, Ramakrishnan et al. suggested that hyperglycemia-induced O-GlcNAcylation of p65 and c-Rel results in excess production of cytokines, thus predisposing cells to cancer development [20,35]. Collectively, these reports not only support the notion that O-GlcNAcylation as a PTM links metabolic changes to inflammation and tumor progression but also affirm that OGT is important for the regulation of NF-κB signaling.

In recent years, the NF-κB signaling pathway has emerged as a prime target to inhibit tumor progression in PDAC. One attractive approach to inhibiting NF-κB is by targeting its regulators. Given the plethora of reports on the significance of p65 site-specific O-GlcNAcylation, it is important to elaborate on the mechanistic knowledge of this PTM in PDAC. Previous studies have demonstrated that O-GlcNAcylation of p65 at T352 and T322 increases anchorage-independent growth in PDAC, with no further evaluation of its molecular impact on NF-κB regulation [13]. Other studies have reported the O-GlcNAcylation of p65 at different sites, such as S319, S337, S374, and T305 [36]. However, O-GlcNAcylation at S337 and S374 does not affect NF-κB activity. Thus, none of the previous studies mechanistically characterized the impact of specific O-GlcNAcylation sites of p65 on NF-κB signaling activity and its pathophysiological functions. Our current study represents the first report on this novel aspect of NF-κB regulation.

In our study, for the first time, we show that S550A, S551A, and S550A/S551A downregulate the majority (~70%) of the NF-κB-dependent genes in PDAC cells. Most of these genes are involved in key oncogenic molecular and cellular functions and are regulated by NF-κB activators (Figure 3). For example, using qPCR, we observed that CD274 (PD-L1) and FGF8 transcripts are downregulated in mutant p65. Among the p65 mutants, we observed that the gene expression pattern in qPCR is not completely consistent with the transcriptional activity pattern observed in the NF-κB luciferase assay (Figure 2). We speculate that the pattern difference between p65 mutants’ transcriptional activity and gene expression is likely due to the differential binding of each mutant to their respective genes’ promoters. CD274 (PD-L1) and FGF8 are known to be associated with poor PDAC prognosis and are important for immune evasion in PDAC, respectively [37,38]. Hence, the O-GlcNAcylation of p65 at S550 and S551 may be implicated in the challenges observed in PDAC immunotherapy endeavors and the overall survival of PDAC patients. More studies will be required to evaluate the functional implications of reduced PD-L1 expression in p65 mutant PDAC cells.

Similarly, our findings have revealed that S550A and S551A mutations may affect the dynamics of one of the major p65’s major phosphorylation sites in the TAD, S536. This observation was made in both PANC1 and MIA PaCa2 cells, as illustrated in Figure 5. The phosphorylation of p65 on S536 is extensively implicated in the promotion of oncogenic processes across various cancer types, and its increased levels are often linked with poor clinical outcomes, as previously reported by Huang et al. [29]. S536 phosphorylation has been shown to facilitate the nuclear translocation of NF-κB and enhance its transactivation via increased binding to CBP/p300, which, in turn, acetylates lysine (K) 310 [23]. Furthermore, the work of Ma et al. in 2013 has shed light on the diminished phosphorylation of p65-S536 upon OGT knockdown in MIA PaCa2 and BxPC-3 cells [13]. This finding suggests a potential complex interplay between O-GlcNAcylation sites on p65 and its phosphorylation sites. Similarly, it is possible that reduced O-GlcNAc at S550 and S551, as modeled in our study, modulates the dynamics of other p65 PTMs, resulting in decreased NF-κB translocation, transcription, and target gene expression. Additional studies will be required to unravel the intricate crosstalk of p65 PTMs and how they work in concert to regulate NF-κB function in PDAC. Nonetheless, considering the challenging nature of targeting certain kinases that phosphorylate S536 [6], the prospect of targeting OGT alone or in combination with kinase inhibitors to attenuate NF-κB activity could serve as a more promising and attractive approach.

Furthermore, it is worth noting that the S550 and S551 residues are evolutionally conserved across multiple species, indicating the significance of the termini serine residues to p65 function (Figure 1E). From a structural perspective, it would be informative to perform in silico modeling to understand the interaction of O-GlcNAcylated p65 at S550 and S551 with notable binding partners. However, the structures of the transactivation domain in the database are disordered. Without these residues being part of a folded structure, it is not possible to predict how the modification would alter the association between the transactivation domain and its interacting partners. It is possible that the O-GlcNAcylation can either stabilize or destabilize these interactions. In addition, for the purposes of this study, we used overexpression of WT and mutant p65 to evaluate the site-specific function of S550 and S551 O-GlcNAcylation in PDAC. Given that knocking down endogenous p65 can hinder cell growth and confound experimental results, we included the empty vector expressing cells in the luciferase assay. It is a well-known phenomenon that overexpression of WT-p65 can dramatically activate NF-κB [8]. From the assay result illustrated in Figure 2, we observed a significantly low level of NF-κB activity in the vector cell lines. In contrast, NF-κB activity was dramatically activated upon overexpression of WT-p65. This suggests the limited contribution of endogenous p65 to NF-κB transcriptional activity and its downstream functions compared to the WT and mutant p65 overexpression cell lines.

## 5. Conclusion

In summary, our study reveals that p65 is O-GlcNAcylated at S550 and S551 and is critical to the effective function of the NF-κB signaling pathway in PDAC cells. For the first time, we have provided extensive mechanistic insights into the role of site-specific O-GlcNAcylation on p65 in PDAC. As a result, we anticipate that our findings will serve as a basis for using OGT as a potential and effective therapeutic target to mitigate the oncogenic impact of O-GlcNAcylated p65 in PDAC patients.

## Figures and Tables

**Figure 1 cancers-15-04742-f001:**
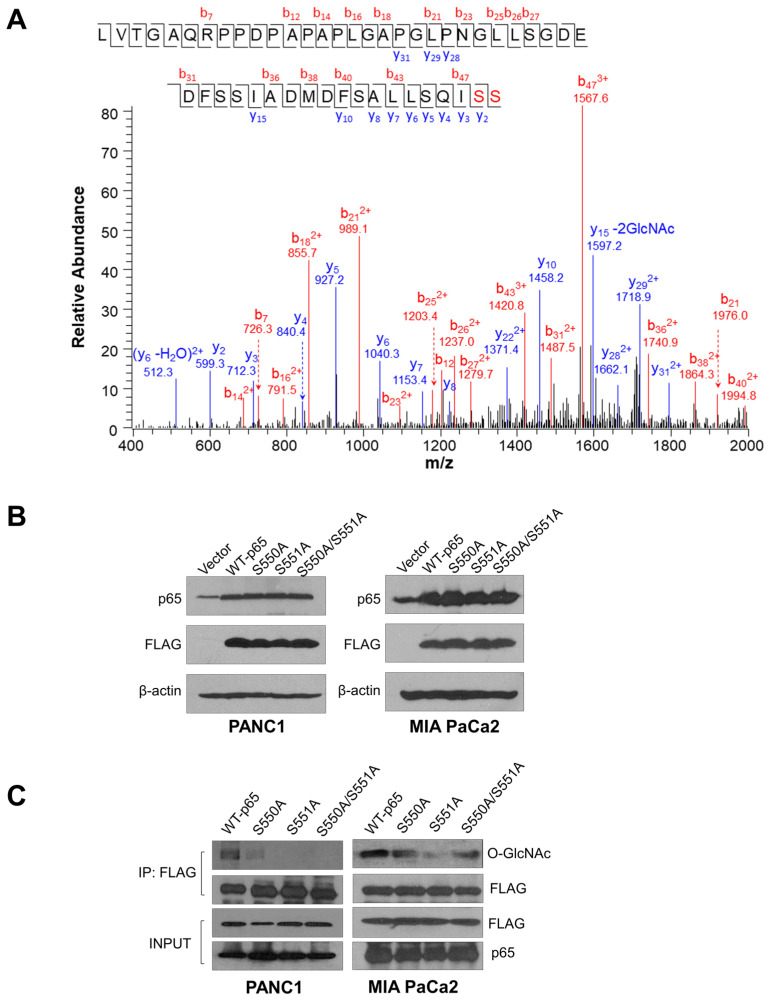
The p65 subunit of NF-κB is O-GlcNAcylated at S550 and S551. (**A**) Tandem mass spectrometry (MS) experiment identifies S550 and S551 on p65 as an O-GlcNAcylated residue with high NF-κB activity. A mass shift of 203.0794 Da was observed for each of the terminal serine residues, indicating the presence of the O-GlcNAcylation modification. (**B**) Generation of FLAG-tagged wild-type (WT)-p65, serine-to-alanine single mutant p65 (S550A and S551A), and double mutant p65 (S550A/S551A) overexpression PANC1 and MIA PaCa2 cells. Western blot images show the overexpression of FLAG-tagged p65 constructs probed with anti-p65, FLAG, or β-actin, respectively, in PANC1 or MIA PaCa2 cells. (**C**) Confirmation of O-GlcNAcylation of p65 at S550 and S551 using co-immunoprecipitation and western blot analysis. Either PANC1 or MIA PaCa2 cells were treated with 50 μM Thiamet G for 24 h. Cell lysates were collected, and FLAG-tagged WT-p65 or mutant p65-S550A, S551A, and S550A/S551A proteins were then immunoprecipitated with anti-FLAG beads and subjected to western analysis using an anti-O-GlcNAc (CTD110.6) antibody. The inputs were probed with anti-FLAG and anti-p65 antibodies. (**D**) Table, showing the list of previously identified p65 GlcNAc sites and their predicted O-GlcNAcylation potential [19,20,21] (https://services.healthtech.dtu.dk/service.php?YinOYang-1.2, accessed on 16 May 2022) [18]. (**E**) Cross-species alignment of amino acid sequences from p65 proteins (residues 510–551). The conserved S550 and S551 residues are indicated in a box (www.uniprot.org, accessed on 21 February 2022). (*) signifies residues aligned across all the species listed.

**Figure 2 cancers-15-04742-f002:**
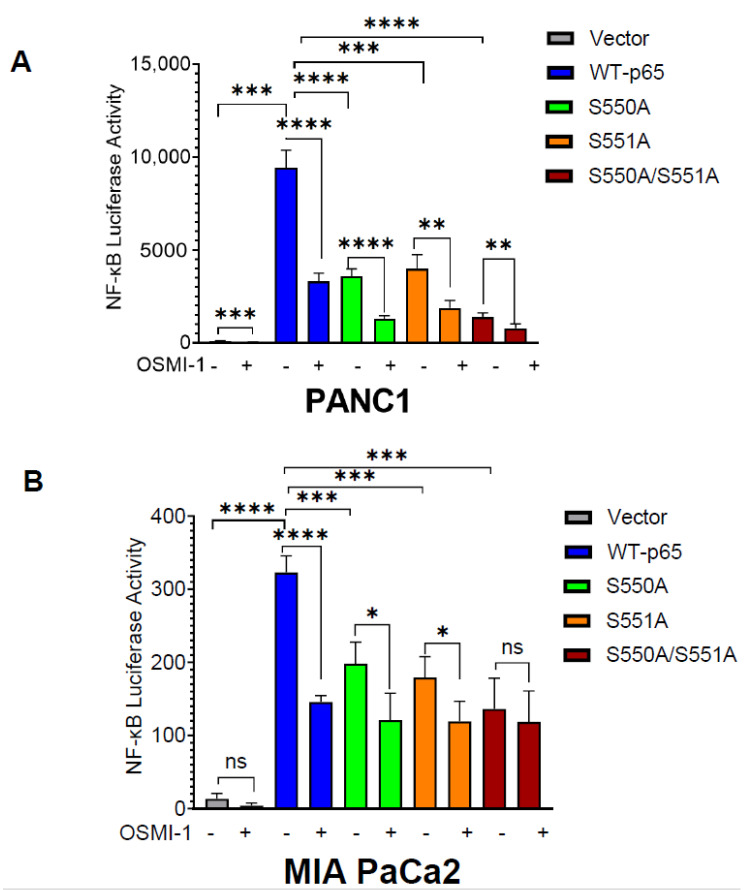
O-GlcNAcylation of p65 at S550 and S551 is important for NF-ĸB activation. (**A**) NF-κB luciferase assay, S550A and S551A showed decreased NF-κB transcriptional activity as compared to WTp65, and 20 μM OSMI-1 treatment decreased NF-κB activity in WT-p65, but not as much in mutant S550A, S551A, and S550A/S551A PANC1 or (**B**) MIA PaCa2 cells. The data represent the means ± standard deviation (S.D.) for three replicates. (*, ***, ****) represents the statistical significance. * *p* < 0.05; ** *p* < 0.001; *** *p* < 0.0002; **** *p* < 0.0001; ns = non-significant.

**Figure 3 cancers-15-04742-f003:**
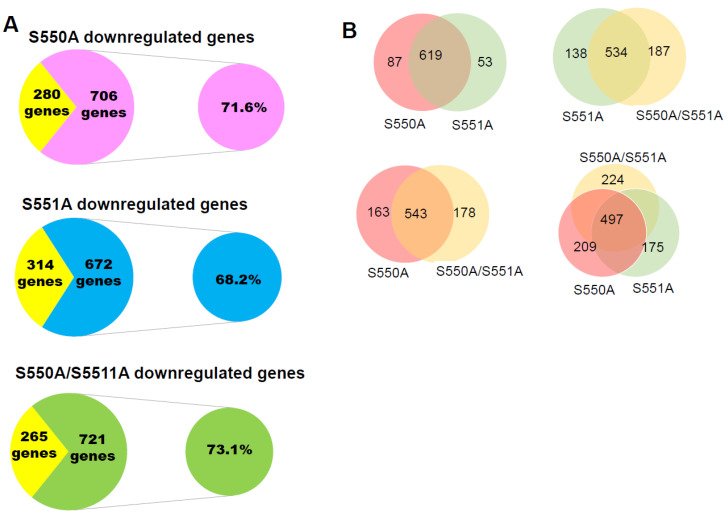
S550A and S551A downregulate a subset of NF-κB target genes. (**A**) Pie chart, representing gene expression data obtained from RNA-seq in PANC1 series of cells. Data shows that of the 986 genes upregulated by 2.5-fold or greater in FLAG-tagged WT-p65 overexpression PANC1 cells, 71.6%, 68.2%, and 73.1% were significantly downregulated in S550A, S551A, and S550A/S551A overexpression PANC1 cells, respectively. Note: Genes that are induced by overexpression of WT-p65 are viewed as NF-κB target genes. (**B**) Venn diagram, showing numbers of commonly downregulated genes between mutant p65-expressing cells. (**C**) Ingenuity Pathway Analysis (IPA): the group of commonly downregulated genes in mutant S550A, S551A, and S550A/S551A cells was used to conduct the IPA. The enrichment results show top upstream regulators, disease and biological functions, molecular and cellular functions, and canonical pathways. (**D**) IPA representative network, showing genes downregulated by mutant S550A, S551A, and S550A/S551A with NF-κB as one of the important nodes in this network. (**E**) Table, showing a short list of NF-κB-dependent genes that are upregulated by WT-p65 and downregulated by the mutant S550A, S551A, and S550A/S551A. (**F**) Validation of RNA-seq data with qPCR, S550A, S551A, and S550A/S551A showed decreased expression of CD274 (PD-L1) and FGF8 in PDAC cells expressing p65 mutants compared to WT-p65. The data represents the means ± standard deviation (S.D.) for three independent experiments. * *p* < 0.05 vs. vector group; # *p* < 0.05 vs. WT-p65 group. The different bar colors represent the distinct cell types labeled on the *x*-axis.

**Figure 4 cancers-15-04742-f004:**
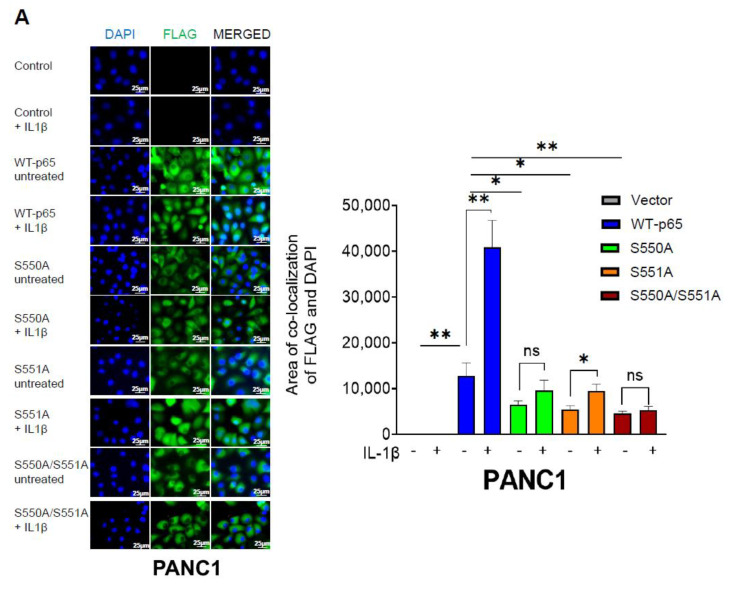
The S550A and S551A mutations of p65 reduce NF-ĸB nuclear translocation and does not affect IĸBα degradation. (**A**,**B**) Left Panel: Immunofluorescence experiment showing localization of FLAG-tagged p65 (green) with an anti-FLAG monoclonal antibody and of nuclei stained with DAPI (blue) and areas of co-localization (cyan) in PANC1 (**A**) and MIA PaCa2 cells (**B**) treated with or without IL-1β (scale bar: 25 μm). FLAG-WT-p65 exhibited increased localization to both nuclei, with more being localized in the nucleus upon IL-1β. Right panel: Quantification of areas of FLAG and DAPI colocalization (cyan), which is representative of FLAG-p65 nuclear translocation. The data represent the means ± standard deviation (S.D.) for three different fields of view. The data represent the means ± standard deviation (S.D.) for three replicates. * *p* < 0.05; ** *p* < 0.001; **** *p* < 0.0001; ns = non-significant (**C**,**D**) Top Panel: Western blot, showing similar IL-1β-induced IĸBα degradation pattern in FLAG-tagged WT-p65, S550A, S551A, and S550A/S551A in PANC1 (**C**) and MIA PaCa2 cells (**D**). Bottom Panel: Densitometry of western blot quantifying total IκBα expression normalized to β-actin. * *p* < 0.05 untreated vs. IL-1β group; n.s. = not statistically significant.

**Figure 5 cancers-15-04742-f005:**
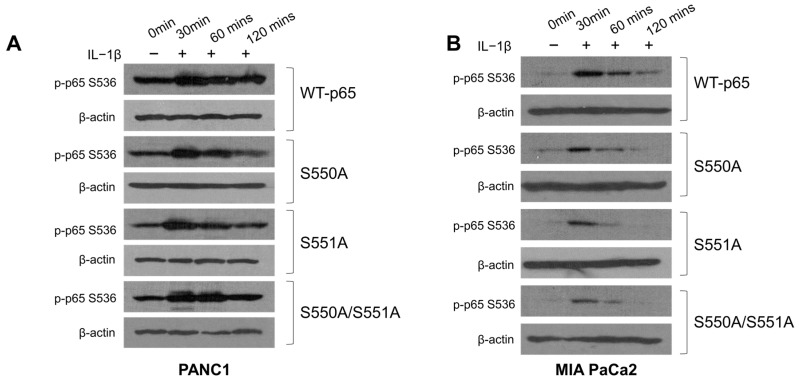
The S550A and S551A mutations of p65 may compromise the strength and duration of S536 phosphorylation in PDAC. (**A**) PANC1 and (**B**) MIA PaCa2 Western blot, showing reduced and less prolonged p65 S536 phosphorylation upon IL-1β treatment in S550A, S551A, and S550A/S551A mutant cells as compared to the WT-p65 cells.

**Figure 6 cancers-15-04742-f006:**
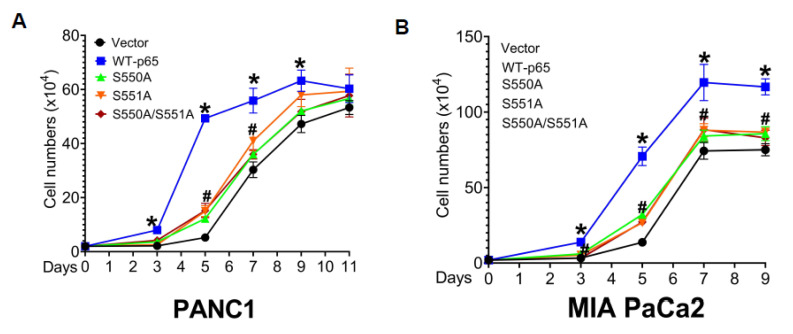
S550A and S551A decrease cell proliferation, 3D spheroid growth, and cell migration in PDAC cells. (**A**,**B**) Cell growth curve, showing cell numbers of vector control (Ctrl), FLAG-tagged WT-p65, S550A, S551A, or S550A/S551A mutant p65 overexpressing cells in PANC1 (**A**) and MIA PaCa2 cells (**B**). Cells (2 × 10^4^) were seeded and counted using a cell counting chamber at days 3, 5, 7, and 9 in MIA PaCa2, including day 11 for PANC1. The data represents the means ± standard deviation (S.D.) for three independent experiments. * *p* < 0.05 vs. vector; # *p* < 0.05 vs. WT. (**C**,**D**) **Top panel:** 3D spheroid growth, representative images of spheroids for PANC1 (**C**) and MIA PaCa2 (**D**) stable cells shown in 4× magnification (scale bar: 25 μm). **Bottom panel:** Quantification of the spheroid growth using fluorescence readings obtained after alamarBlue™ treatment in corresponding cell types. The data represents the means ± standard deviation (SD) for three independent experiments. * *p* < 0.05 vs. Ctrl group; # *p* < 0.05 vs. WT group. (**E**,**F**) **Top panel:** Crystal violet-stained cells representative images (20× magnification), showing the migration of PANC1 (**E**) and MIA PaCa2 (**F**) cells overexpressing FLAG-tagged WT-p65, S550A, S551A, or S550A/S551A mutants compared to the vector control in a Boyden chamber assay. **Bottom panel:** Quantification of the average number of migrated cells is shown (scale bar: 25 μm). The data represents the means ± standard deviation (S.D.) for three independent experiments. * *p* < 0.05 vs. Ctrl group; # *p* < 0.05 vs. WT group.

**Figure 7 cancers-15-04742-f007:**
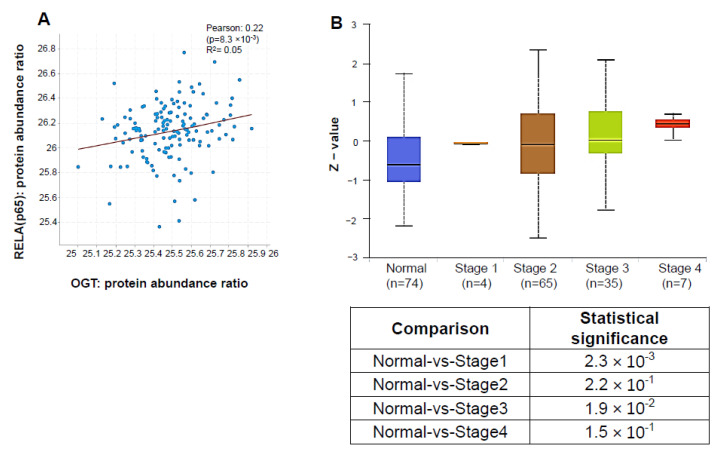
OGT protein abundance positively correlates with p65 and is elevated in PDAC patients. (**A**) Scatter plot showing the positive correlation between OGT and p65 protein abundance ratio based on the Pancreatic Ductal Adenocarcinoma (CPTAC, Cell 2021) study on https://www.cbioportal.org/, accessed on 22 February 2022. (**B**) **Top panel:** Box-whisker plots, illustrating the OGT proteomic expression profile across pancreatic ductal adenocarcinoma (PDAC) tumors and normal pancreatic tissue based on individual cancer stages. The z-values represent standard deviations from the median across samples for the individual cancer stages. **Bottom panel:** The statistical significance between normal and each cancer stage. Individual cancer stages are based on the Clinical Proteomic Tumor Analysis Consortium (CPTAC) dataset (http://ualcan.path.uab.edu, accessed on 22 February 2022).

**Figure 8 cancers-15-04742-f008:**
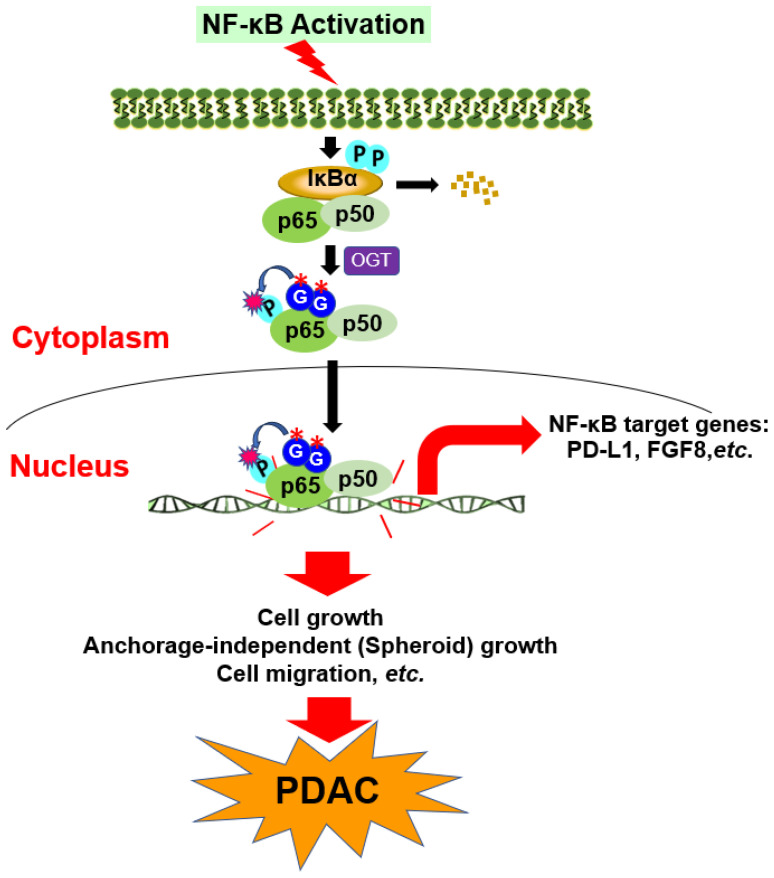
Hypothetical model. In this study, we hypothesize that upon stimulation, the p65 subunit of NF-κB is O-GlcNAc-modified at S550 and S551 by OGT, both of which are critical residues in the transactivation domain of p65. Mechanistically, the p65 O-GlcNAc at S550 and S551 may promote NF-κB nuclear translocation, enhance the strength and duration of p65 S536 phosphorylation, and increase the transcriptional activity independent of IκBα degradation, leading to the increased expression of critical NF-κB target genes such as PD-L1 and FGF8, and so on. Concurrently, these alteration in gene expression promote broader tumor-associated cellular functions such as increasing proliferation, anchorage-independent (3D spheroid) growth, and migration that are induced by p65 overexpression, thus promoting PDAC progression. (*) denotes mutated O-GlcNAc residues.

## Data Availability

Data available upon request.

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
