# Peer review of "Critical Role of Novel O-GlcNAcylation of S550 and S551 on the p65 Subunit of NF-κB in Pancreatic Cancer"

_cancers, 2023, doi:10.3390/cancers15194742_

Round 1
Reviewer 1 Report
This study revealed that p65 is O-GlcNAcylated at S550 and S551 and is critical for the effective functioning of the NF-κB signaling pathway in PDAC cells. It is reported for the first time that p65 undergoes site-specific O-Glc-NAcylation in PDAC. However, there are some questions that need to be addressed.
1. The expression of p65 and the activation model of NF-κB in PDAC cell lines PANC1 and MIA PaCa2 need to be investigated. Additionally, the study should address how to minimize the influence of the basic p65 in this study.
2. In figure 4, it is observed that after the activation of NF-κB, IkappaB levels decreased, but the nuclear translocation of p65 only decreased in p65 mutant cells. Apart from O-Glc-NAcylation, it is important to determine if other post-translational modifications (PTMs) such as phosphorylation and acetylation also changed and require an explanation.
3. There are some vague expressions or inaccurate descriptions in the manuscript which needs to be significantly modified. Additionally, the resolution of the images is not sufficient and requires modification.
The English writing should be improved.
Author Response
Point 1: The expression of p65 and the activation model of NF-κB in PDAC cell lines PANC1 and MIA PaCa2 need to be investigated. Additionally, the study should address how to minimize the influence of the basic p65 in this study.
Response 1: Thanks for your helpful comment. There is a well-known phenomenon in the NF-kB field that overexpression of WT-p65 can activate NF-kB. We have added this information and addressed the point of endogenous p65 in the revised discussion section.
Point 2: In Figure 4, it is observed that after the activation of NF-κB, IkappaB levels decreased, but the nuclear translocation of p65 only decreased in p65 mutant cells.
Response 2: Thanks for your comment. At 1hr, we observed similar levels of decrease of IkBa in both WT and mutant p65 cells, suggesting that the mutation does not impact IkBa degradation. Thus, we called that IkB degradation-independent. However, at the same time point, we observed changes in nuclear translocation, indicating that nuclear translocation is a possible mechanism by which S550 and S551 O-GlcNAcylation modulate NF-kB signaling pathway. We have clarified this information in the revised manuscript.
Point 3: Apart from O-Glc-NAcylation, it is important to determine if other post-translational modifications (PTMs) such as phosphorylation and acetylation also changed and require an explanation.
Response 3: Thank you for this great suggestion. We have assessed NF-κB phosphorylation (phosphorylation of S536 on the p65 of NF-κB) as illustrated in the new Figure 5 and included discussion in the revised Discussion Section.
Point 4: There are some vague expressions or inaccurate descriptions in the manuscript which needs to be significantly modified. Additionally, the resolution of the images is not sufficient and requires modification.
Response 4: We have revised the writing to make it clearer and more specific and the resolution of figures has been greatly increased.
Reviewer 2 Report
Comments:
The manuscript describes " Critical Role of Novel O-GlcNAcylation of S550 and S551 on the p65 subunit of NF-κB in Pancreatic Cancer". This paper shows that overexpression of p65 serine-alanine (S-A) single mutants (S550A or S551A) or double mutants (S550A/S551A) in PDAC cells impairs NF-κB nuclear translocation and transcriptional activity, and p65 mutants downregulate a class of Overexpression of NF-κB target genes that play a role in the maintenance of key cancer features, p65 mutants suppressed cell proliferation, migration, and anchorage-dependent growth of PDAC cells., but several points need to be clarified.
Comment:
1. The resolution of the charts in the article is very poor and should be improved. It is for the benefit of readers to read the article.
2. Do the S550A and S551A mutations affect the phosphorylated IĸBα degradation and the phosphorylated NF-ĸB expression in PANC1 and MIA PaCa2 cells?
3. O-GlcNAcylation of p65 at S550 and S551 is critical for expressing a subset of NF-κB target genes that play an important role in the inflammatory response. Do the S550A and S551A mutations affect the pro-inflammatory cytokines expressions such as TNF-a, IL-Ib and IL-6 in PANC1 and MIA PaCa2 cells?
Minor editing of English language required
Author Response
Point 1: The resolution of the charts in the article is very poor and should be improved. It is for the benefit of readers to read the article.
Response 1: Thanks very much for your valuable comment. We have increased the resolution of the figures.
Point 2: Do the S550A and S551A mutations affect the phosphorylated IĸBα degradation and the phosphorylated NF-ĸB expression in PANC1 and MIA PaCa2 cells?
Response 2: We have included NF-κB phosphorylation (phosphorylation of S536 on the p65 of NF-κB) as illustrated in the new Figure 5 and discussed in the revised discussion section. Also, the degradation of IĸBα is prompted by its phosphorylation. In Figure 4, we show that the degradation pattern of IĸBα is similar across WT and mutant p65 cell lines.
Point 3: O-GlcNAcylation of p65 at S550 and S551 is critical for expressing a subset of NF-κB target genes that play an important role in the inflammatory response. Do the S550A and S551A mutations affect the pro-inflammatory cytokines expressions such as TNF-a, IL-Ib and IL-6 in PANC1 and MIA PaCa2 cells?
Response 3: Thanks for your helpful suggestions, we have added more proinflammatory cytokines obtained from our RNA sequencing data in Figure 3E. The above-mentioned cytokines were not robustly detected in the RNA-seq, thus, they were not included.
Point 4: Writing: minor English editing required.
Response 4: The writing has been thoroughly revised.
Round 2
Reviewer 2 Report
accepted
Minor editing of English language required